# Enhancing Chess Reinforcement Learning with Graph Representation

**Tomas Rigaux**[*]
*Kyoto University*
Kyoto, Japan
tomas@rigaux.com

**Hisashi Kashima**
*Kyoto University*
Kyoto, Japan
kashima@i.kyoto-u.ac.jp

## Abstract

Mastering games is a hard task, as games can be extremely complex, and still fundamentally different in structure from one another. While the AlphaZero algorithm has demonstrated an impressive ability to learn the rules and strategy of a large variety of games, ranging from Go and Chess, to Atari games, its reliance on extensive computational resources and rigid Convolutional Neural Network (CNN) architecture limits its adaptability and scalability. A model trained to play on a $19 \times 19$ Go board cannot be used to play on a smaller $13 \times 13$ board, despite the similarity between the two Go variants. In this paper, we focus on Chess, and explore using a more generic Graph-based Representation of a game state, rather than a grid-based one, to introduce a more general architecture based on Graph Neural Networks (GNN). We also expand the classical Graph Attention Network (GAT) layer to incorporate edge-features, to naturally provide a generic policy output format. Our experiments, performed on smaller networks than the initial AlphaZero paper, show that this new architecture outperforms previous architectures with a similar number of parameters, being able to increase playing strength an order of magnitude faster. We also show that the model, when trained on a smaller $5 \times 5$ variant of chess, is able to be quickly fine-tuned to play on regular $8 \times 8$ chess, suggesting that this approach yields promising generalization abilities. Our code is available at `https://github.com/akulen/AlphaGateau`.

## 1   Introduction

In the past decade, combining Reinforcement Learning (RL) with Deep Neural Networks (DNNs) has proven to be a powerful way to design game agents for a wide range of games. Notable achievements include AlphaGo's dominance in Go [17], AlphaZero's human-like style of play in Chess and Shogi [18], and MuZero's proficiency across various Atari games [15]. They use self-play and Monte Carlo Tree Search (MCTS) [6] to iteratively improve their performance, mirroring the way humans learn through experience, or intuition, and game-tree exploration.

Previous attempts to make RL-based chess engines were unsuccessful as the MCTS exploration requires a precise position heuristic to guide its exploration. Handcrafted heuristics such as the ones used in traditional minimax exhaustive tree searches were too simplistic, and lacked the degree of sophistication that the random tree explorations of MCTS expects to be able to more accurately evaluate a complex chess position. By combining the advances in computing powers with the progress of the field of Deep Learning, AlphaZero was able to provide an adequate heuristic in the form of a Deep Neural Network that was able to learn in symbiosis with the MCTS algorithm to iteratively improve itself.

---

[*]tomas.rigaux.com

38th Conference on Neural Information Processing Systems (NeurIPS 2024).

However, these approaches rely on rigid, game-specific neural network architectures, often representing games states using grid-based data structures, and process them with Convolutional Neural Networks (CNNs), which limits their flexibility and generalization capabilities. For example, a model trained on a standard $19 \times 19$ Go board cannot easily adapt to play on a smaller $13 \times 13$ board without significant changes to its internal structure, manual parameter transfer, and retraining, despite the underlying similarity of the game dynamics. This inflexibility is further compounded by the extensive computational resources required for training these large-scale models from scratch for each specific game or board configuration. If it was possible to make a single model train of various variants of a game, and on various games at the same time, it would be possible to speed up the training by starting to learn the fundamental rules on a simplified and smaller variant of a game, before presenting the model with the more complex version. Similarly, if a model learned all the rules of chess, it could serve as a strong starting point to learn the rules of Shogi, for example.

It could be possible to design a more general architecture for games such as Go, where moves can be mapped one-to-one with the board grid, so that a model could still use CNN layers and handle differently sized boards simultaneously, but this solution is no longer feasible when the moves become more complex, including having to move pieces between squares, or even dropping captured pieces back onto the board in Shogi.

Those moves evoke a graph-like structure, where pieces, initially positioned on squares, are moved to different new squares, following an edge between those two squares, or nodes. As such, it is natural to consider basing an improved model on a graph representation, instead of a grid representation. We explore replacing CNN layers with GNN layers to implement that idea, and more specifically consider in this paper attention-based GNN layers, reflecting how chess players usually remember the structures that the pieces form, and how they interact with each other, instead of remembering where each individual piece is placed, when thinking about a position.

Representing moves as edges in a graph also introduces the possibility to link the output policy size with the number of edges, to make the model able to handle different game variants with different move structures simultaneously. To do so, it becomes important to have edge features as well as node features, as they will be used to compute for each edge the equivalent move logit. As the classical attention GNN layer, the Graph Attention Network [19] (GAT) only defines and updates node-features, we propose a novel modification of the GAT layer, that we call Graph Attention neTwork with Edge features from Attention weight Updates (GATEAU), to introduce edge-features. We also describe the full model architecture integrating the GATEAU layer that can handle differently sized input graphs as AlphaGateau.

Our experimental results demonstrate that this new architecture, when implemented with smaller networks compared to the original AlphaZero, outperforms previous architectures with a similar number of parameters. AlphaGateau exhibits significantly faster learning, achieving a substantial increase in playing strength in a fraction of the training time. Additionally, our approach shows promising generalization capabilities: a model trained on a smaller $5 \times 5$ variant of chess can be quickly fine-tuned to play on the standard $8 \times 8$ chessboard, achieving competitive performance with much less computational effort.

## 2   Related Work

**Reinforcement Learning.** AlphaGo [17], AlphaZero [18], MuZero [15], and others have introduced a powerful framework to exploit Reinforcement Learning techniques in order to generate self-play data used to train from scratch a neural network to play a game.

However, those frameworks use rigid neural networks, that have to be specialized for one specific game. As such, the training process requires a lot of computation resources. It is also not possible to reuse the training on one type of game to train for another one, or to start the training on a smaller and simpler variant of the game, before introducing more complexity.

**Scalable AlphaZero.** In the research of Ben-Assayag and El-Yaniv [2], using Graph Neural Networks has been investigated as a way to solve those issues. Using a GNN based model, it becomes possible to feed as input differently-sized samples, such as Othello boards of size between 5 and 16, enabling the model to learn how to play in a simpler version of the game.

**Algorithm 1:** Self-Play Training

---

**Parameters:** $N_{iter} = 100, N_{games} = 256, N_{sim} = 128, ws = 10^6, N_{train} = 1, bs = 2048$
$\theta \leftarrow$ model.init();
**for** $i \leftarrow 1$ **to** $N_{iter}$ **do**
    data $\leftarrow$ selfplay$(\theta, N_{games}, N_{sim})$ ;               `/* We generate self-play data */`
    frame_window $\leftarrow$ (data||frame_window)$[1:ws]$ ;  `/* The new frame window consists of`
     `the newly generated data and a uniform sample of the previous window */`
    **for** $j \leftarrow 1$ **to** $N_{train}$ **do**
        frame_window $\leftarrow$ frame_window.shuffle();
        **for** batch **in** frame_window.batches$(bs)$ **do**
            $\theta \leftarrow$ apply$(\theta, $gradient$(\theta, $batch$))$;
        **end**
    **end**
**end**

---

This approach had promising results, with 10 times faster training time than the AlphaZero baseline. It was however limited to Othello and Gomoku, and using the GNN layers (GIN layers [20]) only as a scalable variant of CNN layers, keeping a rigid grid structure.

**Edge-featured GNNs.** There exists a large variety of GNN variants, specialized for different use case s and data properties. For this work, a simple layer was enough to experiment with the merits of the proposed approach, except it was critical that the chosen layer handled both node-features and edge-features. We chose to use an attention-based layer.

Gong and Cheng [10] introduce the EGNN(a) layer, where each dimension of an edge-feature is used for a different attention head. We wanted edge features to be treated as a closer equivalent to node-features, so we did not use this layer.

The EGAT layer introduced by Chen and Chen [4] is better for our case, as they construct a dual graph where edges and nodes have reversed roles, so the node features in the dual graph are edge features for the initial graph. However, this method requires building the dual graph, and is quadratic in the maximal node degree. As this was quite complex, we decided to introduce GATEAU, which solves the problem in a simpler and more natural way.

## 3 Setting

Our architecture is based on the AlphaZero framework, which employs a neural network $f_\theta$ with parameters $\theta$ that is used as an oracle for a Monte-Carlo Tree Search (MCTS) algorithm to generate self-play games. When given a board state $s$, the neural network predicts a (value, policy) pair $(v(s), \pi(s, \cdot)) = f_\theta(s)$, where the value $v \in [-1, 1]$ is the expected outcome of the game, and the policy $\pi$ is a probability distribution over the moves.

We utilize Algorithm 1 to train the models in this paper, with modifications to incorporate Gumbel MuZero [7] with a gumbel scale of 1.0 as our MCTS variant.

## 4 Proposed Models

### 4.1 Motivation: Representing the Game State as a Graph

Many games, including chess, are not best represented as a grid. For example, chess moves are analogous to edges in a grid graph, and games like Risk naturally form planar graphs based on the map. As such, it makes natural sense to encode more information through graphs in the neural network layers that are part of the model.

This research focuses on implementing this idea in the context of chess. This requires to answer two questions: how to represent a chess position as a graph, and how to output a policy vector that is edge-based, and not node-based.

The architecture presented in this paper is based on GNNs, but using node features to evaluate the value head, and edge features to evaluate the policy head. As such, a GNN layer that is able to handle

**Table 1:** Node features

| Dimensions | Description |
|---|---|
| 12 | The piece on the square, as a 12-dimensional soft one-hot |
| 2 | Whether the position was repeated before |
| 98 | For each of the previous 7 moves, we repeat the last 14 dimensions to describe the corresponding positions |
| 1 | The current player |
| 1 | The total move count |
| 4 | Castling rights for each player |
| 1 | The number of moves with no progress |

**Table 2:** Edge features

| Dimensions | Description |
|---|---|
| 1 | Is this move legal in the current position? |
| 2 | How many squares {to the left/up} does this edge move? |
| 4 | Would a pawn promote to a {knight/bishop/rook/queen} if it did this move? |
| 2 | Could a {white/black} pawn do this move? |
| 4 | Could a {knight/bishop/rook/queen} do this move? |
| 2 | Could a {white/black} king do this move? |

both node and edge features is required. This paper will introduce the GATEAU layer, that is a natural extension of the GAT layer [19] to edge features.

## 4.2 Graph Design

In AlphaZero, a chess position is encoded as an $n \times n \times 119$ matrix, where each square on the $n \times n$ chess board is associated to a feature vector of size 119, containing information about the corresponding square for the current position, as well as the last 7 positions, as described in Table 1.

We will instead represent the board state as a graph $G(V, E)$, with the $n \times n$ squares being nodes $V$, and the edges $E$ being moves, based on the action design of AlphaZero. Each AlphaZero action is a pair (source square, move), with $n \times n$ possible source squares. In $8 \times 8$ chess, the 73 moves (resp. 49 in $5 \times 5$ chess) are divided into 56 queen moves (resp. 32), 8 knight moves (resp. 8), and 9 underpromotions (resp. 9) for a total of 4672 actions (resp. 1225). The edge associated with an action is oriented from the node corresponding to the source square, to the destination square of the associated move. In $8 \times 8$ chess, castling is represented with the action going from the king's starting square going laterally 2 squares. As this action encoding is a little too large, containing moves ending outside of the board that do not correspond to real edges, the constructed graph only contains 1858 edges (resp. 455), corresponding only to valid moves.

The node and edge features, of initial size 119 and 15, are detailed in Tables 1 and 2, respectively. Node features are based on AlphaZero's features, including piece type, game state information, and historical move data. Edge features encode move legality, direction, potential promotions, and piece-specific move capabilities. In the case of $5 \times 5$ chess, we include the unused castling information, in order to have the same vector size of the $8 \times 8$ models. It would be possible to preprocess the node and edge features differently for different games or variants, but for simplicity we didn't do it.

The starting positions for all games played in our experiments were either the classical board setup in $8 \times 8$ chess, or the Gardner setup for $5 \times 5$ chess, illustrated in Figure 1.

## 4.3 GATEAU: A New GNN Layer, with Edge Features

The Graph Attention Network layer introduced by Veličković et al. [19] updates the node features by averaging the features of the neighbouring nodes, weighted by some attention coefficient. To be more precise, given the node features $h \in \mathbb{R}^{N \times K}$, attention coefficients are defined as

$$e_{ij} = W_u h_i + W_v h_j \tag{1}$$

with parameters $W_u, W_v \in \mathbb{R}^{K \times K'}$. In the original paper, $W_u = W_v$, but as we are working with a directed graph, we differentiate them to treat the source and destination node asymmetrically. Then

we can compute attention weights $\alpha$, and use them to update the node features:

$$\alpha_{ij}^0 = \text{softmax}_j\left(\text{LeakyReLU}\left(a^T e_{i.}\right)\right) = \frac{\exp^{\text{LeakyReLU}\left(a^T e_{ij}\right)}}{\sum_k \exp^{\text{LeakyReLU}\left(a^T e_{ik}\right)}}, \tag{2}$$

$$h_i' = \sum_{j \in \mathcal{N}_i} \alpha_{ij}^0 W h_j \tag{3}$$

with parameters $W \in \mathbb{R}^{K \times K''}$ and $a \in \mathbb{R}^{K'}$.

The main observation motivating GATEAU is that in this process, the attention coefficients $e_{ij}$ serve a role similar to node features, being a vector encoding some information between nodes $i$ and $j$. As such, we propose to introduce edge features in place of those attention coefficients.

Our proposed layer, called Graph Attention neTwork with Edge features from Attention weight Updates (GATEAU) takes the node features $h \in \mathbb{R}^{N \times K}$ and edge features $g_{i,j} \in \mathbb{R}^{N \times N \times K'}$ as inputs. We start by simply updating the edge features similarly to Eq. 1:

$$g_{ij}' = W_u h_i + \underline{W_e g_{ij}} + W_v h_j \tag{4}$$

with parameters $W_u, W_v \in \mathbb{R}^{K \times K'}$ and $W_e \in \mathbb{R}^{K' \times K'}$. Then the attention weights are obtained as in Eq. 2, by substituting the attention coefficients with our new edge features:

$$\alpha_{ij} = \text{softmax}_j\left(\text{LeakyReLU}\left(a^T g_{i.}'\right)\right) \tag{5}$$

with parameter $a \in \mathbb{R}^{K'}$. Finally, we update the node features as in Eq. 3:

$$h_i' = W_0 h_i + \sum_{j \in \mathcal{N}_i} \alpha_{ij}\left(W_h h_j + W_g g_{ij}\right) \tag{6}$$

with parameters $W_0, W_h \in \mathbb{R}^{K \times K''}$ and $W_g \in \mathbb{R}^{K' \times K''}$. We add the self-edges manually as it is inconvenient for the policy head if they are included in the graph, and we mix back the values of the edge features back in the node features.

### 4.4 AlphaGateau: A Full Model Architecture Based on AlphaZero and GATEAU

Following the structure of the AlphaZero neural network, we introduce AlphaGateau, combining AlphaZero with GATEAU instead of CNN layers, and redesign the value and policy head to be able to exploit node and edge features respectively to handle arbitrarily sized inputs with the same number of parameters.

We define the following layers, which are used to describe AlphaGateau in Figure 2.

**Attention Pooling.** In order to compute a value for a given graph, we need to pool the features together. Node features seem to be the more closely related to positional information, so we pool them instead of edge features. For this, we use an attention-based pooling layer, similar to the one described in Eq. 7 by Li et al. [14], which, for node features $h \in \mathbb{R}^{N \times K}$ and a parameter vector $a \in \mathbb{R}^K$, outputs

$$\alpha_i^p = \text{softmax}_i\left(\text{LeakyReLU}\left(a^T h_{.}\right)\right),$$
$$H = \sum_i \alpha_i^p h_i, \tag{7}$$

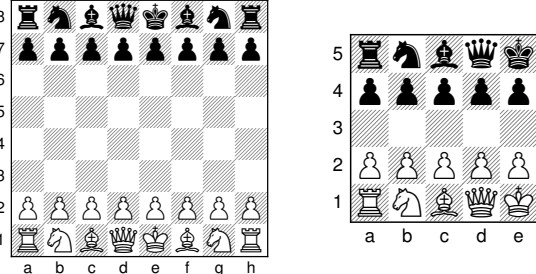

**Figure 1:** The starting positions of $8 \times 8$ and $5 \times 5$ chess games

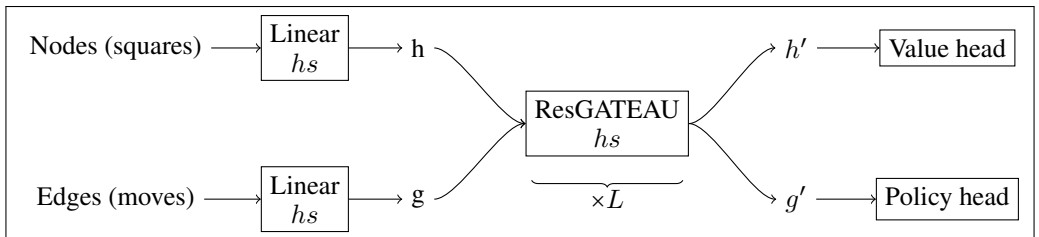

**Figure 2:** The AlphaGateau network, $hs$ is the inner size of the feature vectors, and $L$ is the number of residual blocks.

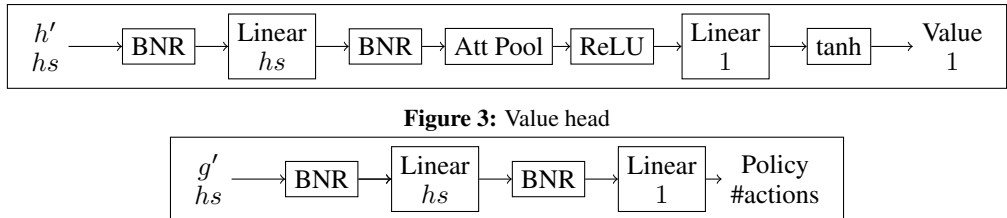

**Figure 3:** Value head

**Figure 4:** Policy head

where $H \in \mathbb{R}^K$ is a global feature vector.

**Batch Normalization and Non-linearity (BNR).** As they are a pair of operations that often occur, we group Batch Normalization and a ReLU layer together under the notation BNR:

$$\mathrm{BNR}(x) = \mathrm{ReLU}(\mathrm{BatchNorm}(x)). \tag{8}$$

**Residual GATEAU (ResGATEAU).** Mirroring the AlphaZero residual block architecture, we introduce ResGATEAU, which similarly sums a normalized output from two stacked GATEAU layers to the input:

$$\mathrm{ResGATEAU}(h,g) = (h,g) + \mathrm{GATEAU}(\mathrm{BNR}(\mathrm{GATEAU}(\mathrm{BNR}(h,g)))). \tag{9}$$

# 5 Experiments

We evaluate AlphaGateau's performance in learning regular $8 \times 8$ chess from scratch and generalizing from a $5 \times 5$ variant to the standard $8 \times 8$ chessboard. The metric used to evaluate the models is the Elo rating, calculated through games played against other models (or players) with similar ratings. Due to computational constraints, we couldn't replicate the full 40 residual layers used in the initial AlphaZero paper, and experimented with 5 and 6 layer models. We also started exploring 8 layers, but these models required to generate a lot more data, which would make the experiment run an order of magnitude longer. Our results indicate that AlphaGateau learns significantly faster than a traditional CNN-based model with similar structure and depth, and can be efficiently fine-tuned from $5 \times 5$ to $8 \times 8$ chess, achieving competitive performance with fewer training iterations. [2] All models used in these experiments are trained with the Adam optimizer [12] with a learning rate of 0.001. All feature vectors have an embedding dimension of 128. The loss function is the same as for the original AlphaZero, which is, for $f_\theta(s) = \tilde{\pi}, \tilde{v}$,

$$L(\pi, v, \tilde{\pi}, \tilde{v}) = -\pi^T \log(\tilde{\pi}) + (v - \tilde{v})^2. \tag{10}$$

---

[2]In Silver et al. [18], the training of AlphaZero is described in terms of steps, which each consists of one mini-batch of size 4096, while the generation of games through self-play is done in parallel by other TPUs. For our experiments, an iteration consists of generating 256 games through self-play, then doing one epoch of training, split into 3904 mini-batches of size 256, after 7 iterations once the frame window is full. In terms of positions seen, one iteration is equivalent to 244 steps.

## 5.1 Implementation

**Jax and PGX.** As the MCTS algorithm requires a lot of model calls weaved throughout the tree exploration, it is essential to have optimized GPU code running both the model calls, and the MCTS search. In order to leverage the MCTX [8] implementations of Gumbel MuZero, all our models and experiments were implemented in Jax [3] and its ecosystem [11] [9]. PGX [13] was used for the chess implementation, and we based our AlphaZero implementation on the PGX implementation of AZNet. We used Aim [1] to log all our experiments.

To estimate the Elo ratings, we use the statsmodels package [16] implementation of Weighted Least Squares (WLS).

**Hardware.** All our models were trained using multiple Nvidia RTX A5000 GPUs (*Learning speed* used 8 and *Fine-tuning* used 6), and their Elo ratings were estimated using 6 of those GPUs.

## 5.2 Evaluation

As each training and evaluation lasted a little under a week, we were not able to train each model configuration several times so far. As such, each model presented in the results was trained only once, and the confidence intervals that we include are on the Elo rating that we estimated for each of them, as described in the following.

During training, at regular intervals (each 2, 5, or 10 iterations), the model parameters were saved, and we used this dataset of parameters to evaluate Elo ratings. In this section, we will call a pair (model, parameters) a player, and compute a rating for every player.

We initially chose 10 players, and simulated 60 games between each pair of players, to get initial match data $M$. For each pair of players that played a match, we store the number of wins $w_{ij}$, draws $d_{ij}$, and losses $l_{ij}$: $M_{ij} = (w_{ij}, d_{ij}, l_{ij})$. Using this data, we can roughly estimate the ratings $r \in \mathbb{R}^{N_{players}}$ of the players present in $M$ using a linear regression on the following set of equations:

$$
\begin{cases}
r_j - r_i & = \dfrac{400}{\log(10)} \log \left( \dfrac{w_{ij} + d_{ij} + l_{ij} + 1}{w_{ij} + \frac{d_{ij}+1}{2}} - 1 \right) \quad \text{for } i \in M, j \in M_i, \\
\sum_{i \in M} r_i & = |M| \times 1000.
\end{cases}
\tag{11}
$$

We artificially add one draw to avoid extreme cases where there are only wins for one player and no losses, in which case the rating difference would theoretically be infinite. This is equivalent to a Jeffreys prior. The last equation fixes the average rating to 1000, as the Elo ratings are collectively invariant by translation.

We then ran Algorithm 2 to generate a densely connected match data graph, where each player played against at least 5 other players of similar playing strength. Finally, we used this dataset to fit a linear regression model (Weighted Least Squares) to get Elo ratings that we used in the results figures for the experiments. The confidence intervals were estimated by assuming that the normalized match outcomes followed a Gaussian distribution. If $\hat{p}_{ij} = \frac{w_{ij} + \frac{d_{ij}+1}{2}}{w_{ij} + d_{ij} + l_{ij} + 1}$ is the estimated probability that player $i$ beats player $j$, we approximate the distribution that $p_{ij}$ follows as a Gaussian, and using the delta method, we derive that $r_j - r_j$ asymptotically follows a Gaussian distribution of mean $\frac{400}{\log(10)} \log \left( \frac{1}{p_{ij}} - 1 \right)$ and variance $\left( \frac{400}{\log(10)} \right)^2 \frac{1}{(w_{ij}+d_{ij}+l_{ij})p_{ij}(1-p_{ij})}$. The proof is detailed in Appendix A.2. Using the WLS linear model of statsmodels [16], we get Elo ratings for every player, as well as their standard deviations, which we use in the following to derive 2-sigma confidence intervals.

## 5.3 Experiments

**Learning Speed.** Our first experiment compares the baseline ability of AlphaGateau to learn how to play $8 \times 8$ chess from scratch, and compares it with a scaled down AlphaZero model. The AlphaZero model has 5 residual layers (containing 10 CNN layers) and a total of 2.2M parameters, and the AlphaGateau model also has 5 ResGATEAU layers (containing 10 GATEAU layers) and a total of 1.0M parameters, as it doesn't need an $N_{nodes} \times hs \times N_{actions}$ fully connected layer in the policy head, and the GATEAU layers use 2/3 of the parameters a $3 \times 3$ CNN does.

**Algorithm 2:** Matching Players

---

**Parameters:** $N_{games} = 60, N_{sim} = 128$
**Input:** $M$
**for** player **in** *unmatched players* **do**
    **for** $i \leftarrow 1$ **to** $5$ **do**
        $r \leftarrow \texttt{Elo\_LR}(M)$ ;               `/* The Linear Regression is run on Eq. 11 */`
        opponent $\leftarrow \arg\min_{j \in M} |r_j - r_{\mathsf{player}}|$ ;     `/* If player ∉ M, we set` $r_{\mathsf{player}} = 1000$ `*/`
        $(w, d, l) \leftarrow \texttt{play}(\mathsf{player}, \mathsf{opponent}, N_{games}, N_{sim})$;
        $M_{\mathsf{player,opponent}} \leftarrow M_{\mathsf{player,opponent}} + (w, d, l)$;
        $M_{\mathsf{opponent,player}} \leftarrow M_{\mathsf{opponent,player}} + (l, d, w)$;
    **end**
**end**

---

For this experiment, we generated 256 games of length 512 at each iteration, totalling $131\,072$ frames, and kept a frame window of 1M frames (all the newly generated frames, and uniform sampling over the frame window of the previous iteration), over 500 iterations. During the neural network training, we used a batch size of 256. The training for AlphaGateau lasted 13 days and 16 hours, while AlphaZero took 10 days and 3 hours.

We report the estimated Elo ratings with a 2-sigma confidence interval in Figure 5. AlphaZero was only able to reach an Elo of $667 \pm 38$ in 500 iterations, and would likely continue to improve with more time, while AlphaGateau reached an Elo of $2105 \pm 42$, with an explosive first 50 iterations, and achieving results comparable to the final Elo of AlphaZero after only 10 iterations.

Although those results are promising, it is important to note that we only compared to a simplified version of AlphaZero, using only 5 layers instead of the original 40, and without spending large efforts to optimize the hyperparameters of the model. As such, it is possible that the performance of AlphaZero could be greatly improved in this context with more parameter engineering. Both AlphaGateau and AlphaZero have not reached a performance plateau after 500 iterations, showing slow but consistent growth.

**Fine-tuning.** In our second experiment, we trained a AlphaGateau model with 10 residual layers on $5 \times 5$ chess for 100 iterations, then fine-tuned this model on $8 \times 8$ chess for 100 iterations. This model has a total of 1.7M parameters.

For this experiment, we generated 1024 games of length 256 at each iteration on $5 \times 5$ chess, and 256 games of length 512 while fine-tuning on $8 \times 8$ chess, and kept a frame window of 1M frames. The initial training lasted 2 days and 7 hours, and the fine-tuning 5 days and 15 hours.

We report the estimated Elo ratings with a 2-sigma confidence interval in Figure 6. The initial training ended with an Elo rating of $807 \pm 46$ when evaluated on $8 \times 8$ chess games, which suggests that it was able to learn general rules of chess on $5 \times 5$, and apply them with some success on $8 \times 8$ chess without having seen any $8 \times 8$ chess position during its training. Once the fine-tuning starts, the model jumps to an Elo of $1181 \pm 50$ after a couple iterations, suggesting the baseline learned on $5 \times 5$ was of high quality. After fine-tuning, the model had an Elo of $1876 \pm 47$, reaching comparable performances to the smaller model using roughly the same amount of iterations and GPU-time, despite being twice as big. Preliminary testing suggests that this model would become stronger if more data was generated at each iteration, but that would linearly increase the training time, as generating self-play games took half of the 5 days of training.

## 5.4 Impact of the Frame Window and the Number of Self-play Games

In order to train deeper networks, we experimented with the number of self-play games generated at each generation, and with the size of the frame window. It seems from our results in Figure 7 that having more newly generated data helps the model learn faster. However, the time taken for each iteration scales linearly with the number of generated games, and the model is still able to improve using older data. As such, keeping a portion of the frame window from previous iterations makes for a good compromise. There are however options to improve our frame window selection:

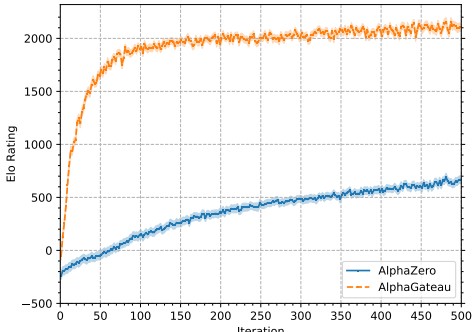 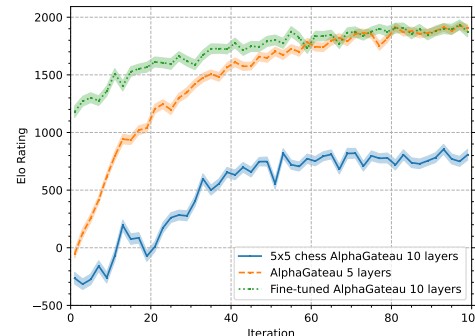

**Figure 5:** The Elo ratings of AlphaZero and Alpha-Gateau with 5 residual layers trained over 500 iterations. The AlphaGateau model initially learns ~10 times faster than the AlphaZero model, and settles after 100 iterations to a comparable speed of growth to that of AlphaZero.

**Figure 6:** The Elo ratings of the first 100 iterations of the AlphaGateau model from Figure 5 was included for comparison. The initial training on $5 \times 5$ chess is able to increase its rating while evaluated on $8 \times 8$ chess during training, even without seeing any $8 \times 8$ chess position. The fine-tuned model starts with a good baseline, and reaches comparable performances to the 5-layer model despite being undertrained for its size.

- Which previous samples should be selected? We selected uniformly at random from the previous frame window to complement the newly generated samples, but it might be preferable to select fewer samples, but chosen as to represent a wide range of different positions.

- Past samples are by design of dubious quality. As they come from self-play games with previous model parameters, they correspond to games played with a lower playing strength, and the policy output by the MCTS is also worse. Keeping a sample that is too old might cause a drop of performance rather than help the model learn. We experimented a little with keeping the 1M most recent samples, but with little success.

We also initially tried to increase the number of epochs in one iteration, but only saw marginal gains, suggesting that mixing new data among the previous frame window helps the model extract more training information from previous samples.

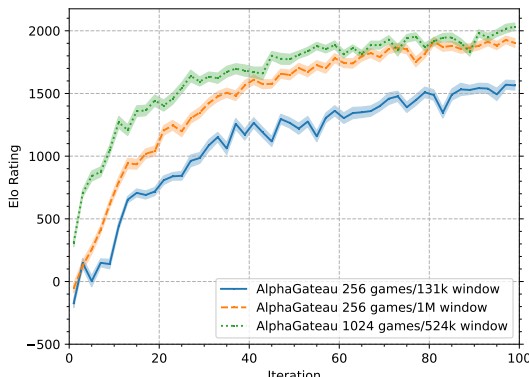

**Figure 7:** The two models with a frame window of size $131\,072$ only kept the latest generated games in the frame window. The model keeping no frame window and generating 256 games was trained in only 39 hours, but had the worst performance. Adding a 1M frame window improved the performance a little and lasted 60 hours, while increasing the number of self-play games to 1024 performed the best, but took 198 hours.

# 6 Conclusion

In this paper, we introduce AlphaGateau, a variant on the AlphaZero model design that represents a chess game state as a graph, in order to improve performance and enable it to better generalize to variants of a game. This change from grid-based CNN to graph-based GNN yields impressive increase in performance, and seems promising to enable more research on reinforcement-learning based game-playing agent research, as it reduces the resources required to train one.

We also introduce a variant of GAT, GATEAU, that we designed in order to handle edge features in a simple manner performed well, and efficiently.

**Future Work.** As our models were relatively shallow when compared to the initial AlphaZero, it would be important to confirm that AlphaGateau still outperforms AlphaZero when both are trained with a full 40-deep architecture. This will require a lot more computing time and resources.

As discussed in Section 5.4, our design of the frame window is a little unsatisfactory, and a future improvement would be to define an efficiently computable similarity metric between chess positions, that helps the neural network generalize.

We focused on chess for this paper, but there are other games that could benefit from this new approach. The first one would be shogi, as it has similar rules to chess, and the promising generalization results from AlphaGateau could be used to either train a model on one game, and fine-tune it on the other, or to jointly train it on both games, to have a more generalized game-playing agent. As alluded to in the Graph Design 4.2, more features engineering would be required to have node and edge features compatibility between chess graphs, and shogi graphs. It could also be possible to change the model architecture to handle games with more challenging properties, such as the game Risk, which has more than 2 players, randomness, hidden information, and varying maps, but is even more suited to being represented as a graph.

## Acknowledgments and Disclosure of Funding

This work was supported by JST BOOST, Grant Number JPMJBS2407 and JST CREST, Grant Number JPMJCR21D1. We thank Armand Barbot and Jill-Jênn Vie for their helpful comments and insightful discussions.

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

# A    Appendix / supplemental material

## A.1    Elo

The Elo rating system was initially introduced as a way to assign a value to the playing strength of chess players. The rating of each player is supposed to be dynamic and be adjusted after each game they play to follow their evolution.

The Elo ratings are defined to respect the following property: If two players with Elo rating $R_A$ and $R_B$ played a game, the probability that player $A$ wins is

$$E_A = \frac{1}{1 + 10^{\frac{R_B - R_A}{400}}}. \tag{12}$$

## A.2    Variance of estimated Elo rating difference

We assume that the outcome of a game between two players of Elo $R_A$ and $R_B$ is a Bernoulli trial, with a probability that player $A$ win being given by the central Elo equation 12. If we want to estimate the Elo of both players, we need to estimate that probability $p$. To do so, we can make the two players play $n$ games, and sum the wins of player $A$, as well as half his draws, to get $x = w + \frac{d}{2}$. From this, we can estimate the value of $p$ using the estimator $\hat{p} = \frac{x}{n}$. In the case that one of the two players is significantly stronger than the other, $\hat{p}$ could be close to $0$ or $1$, in which case this estimator is wildly inaccurate. To remedy this, we will instead rely on a Jeffreys prior, to get the estimator $\hat{p}_{Jeffreys} = \frac{x+1/2}{n+1}$. We will note this estimator $\hat{p}$ in the following.

From our assumptions, we have that $x$ follows a binomial distribution $B(n, p)$, and we will approximate the distribution that $\hat{p}$ follows by a normal distribution $\hat{p} \sim \mathcal{N}(p, \frac{p(1-p)}{n})$.

By inverting the Elo equation 12, we can get the rating difference from the probability that $A$ wins as

$$R_B - R_A = \frac{400}{\log(10)} \log\left(\frac{1}{p} - 1\right), . \tag{13}$$

Therefore, posing $g(y) = \frac{400}{\log(10)} \log\left(\frac{1}{y} - 1\right)$, which is differentiable, we can use the delta method to get that $g(\hat{p})$ is asymptotically Gaussian. The derivative of $g$ is

$$\begin{aligned}
g'(y) &= \frac{400}{\log(10)} \left(-\frac{1}{y^2}\right)\left(\frac{1}{\frac{1}{y} - 1}\right) \\
&= -\frac{400}{\log(10)} \frac{1}{y^2} \frac{y}{y - 1} \\
&= -\frac{400}{\log(10)} \frac{1}{y(y-1)}, \tag{14}
\end{aligned}$$

which gives

$$\begin{aligned}
\hat{R}_B - \hat{R}_A = g(\hat{p}) &\sim \mathcal{N}\left(g(p), \frac{p(1-p)}{n}(g'(p))^2\right) \\
&\sim \mathcal{N}\left(\frac{400}{\log(10)} \log\left(\frac{1}{p} - 1\right), \frac{p(1-p)}{n}\left(-\frac{400}{\log(10)} \frac{1}{p(p-1)}\right)^2\right) \\
&\sim \mathcal{N}\left(\frac{400}{\log(10)} \log\left(\frac{1}{p} - 1\right), \frac{400^2}{\log(10)^2} \frac{p(1-p)}{n} \frac{1}{(p(p-1))^2}\right) \\
&\sim \mathcal{N}\left(\frac{400}{\log(10)} \log\left(\frac{1}{p} - 1\right), \frac{400^2}{\log(10)^2} \frac{1}{np(p-1)}\right). \tag{15}
\end{aligned}$$

## A.3    Comparison with BayesElo

We also used BayesElo [5] to evaluate the Elo ratings of our models using all the PGN files that were generated recording the games played between all our models. In Figure 8, we plotted a point

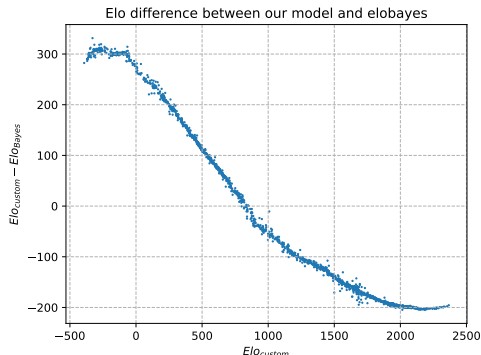

**Figure 8:** The difference between BayesElo ratings and the Elo ratings according to our method. We removed to each Elo the average Elo of all players in its respective method, such that the average effective Elo for both BayesElo and our method is 0

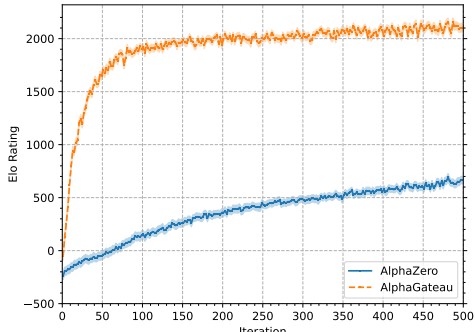

**Figure 9:** A copy of Figure 5 for comparison.

**Figure 10:** Using running time instead of iteration for Figure 5 doesn't change much, as AlphaZero is only a little bit faster than AlphaGateau.

on $(x, y)$ for each player where $x$ is its Elo rating following our method and $y$ is the difference in predicted Elo between our method and BayesElo.

In practice, weaker models tend to be overrated by our method compared with BayesElo, while stronger models are underrated. As this relation is smooth and monotone, this suggests that both methods order the relative strength of all players similarly, with small differences only being due to noise. The main difference being that our range of Elo ratings is compacted towards the extremes, which we assume is due to our practical Jeffreys prior, that implies that strong models drew at least one game against all their opponents, handicapping their Elo, and similarly weak models always manage to not lose at least one game.

## A.4 FLOPs comparison

We didn't record the FLOPs of our models, however, we did record the time taken for each iteration. In Figure 10, 12, and 14, we plotted the three main plots of the main paper using that data on the X-axis. As some experiments were run using 6 GPUs instead of *, we multiplied their recorded time by a factor $\frac{6}{8}$ for better comparison.

## A.5 PGNs

We include with Figure 15, 16, and 17 a few PGNs showing games played by our models.

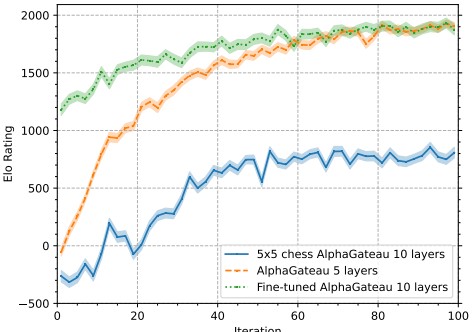

**Figure 11:** A copy of Figure 6 for comparison.

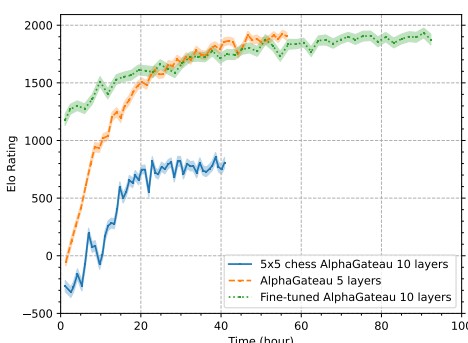

**Figure 12:** Using running time instead of iteration for Figure 5. Training the deeper model takes roughly 40 hours longer, for a similar amount of generated games and training steps.

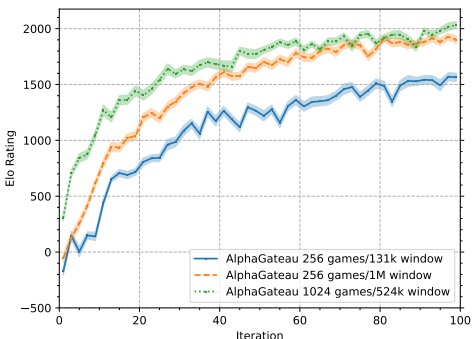

**Figure 13:** A copy of Figure 7 for comparison.

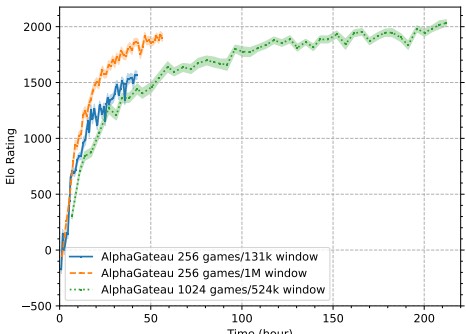

**Figure 14:** Using running time instead of iteration for Figure 5 shows that although using more newly generated games instead of relying on a frame window of previous data makes the model improve more per iteration, it does result in a slower training in practice.

Figure 15 contains the PGN of a game played on $8 \times 8$ chess by the last iteration of the $5 \times 5$ AlphaGateau model of our fine-tuning experiment, to showcase its playing style while having never seen an $8 \times 8$ board during its training.

Figure 16 and 17 contains the PGNs of games played by the last iteration (iteration 499) of the full AlphaGateau model of our first experiment as white and black respectively, selected among the games played against other models to evaluate all the Elo rankings, selected using the script book.py in the GitHub repo, following the most played move each ply. The game played as white in Figure 16 is played against the iteration 61 of a fine-tuned AlphaGateau model using 8 layers, 1024 generated games per iteration, but only 32 MCTS simulations, with an estimated Elo rating of 2124. The game played as black in Figure 17 is played against the iteration 439 of the same model.

### A.6 Lichess Evaluation

We ran the final iteration of the 5-layer AlphaGateau model on Lichess, as the bot AlphaGateau (`https://lichess.org/@/AlphaGateau`). We let it play against other bots and some human players in bullet, blitz, and rapid time formats by varying the number of MCTS simulations to adjust the time required to play each move. This was implemented using the default Lichess bot bridge (`https://github.com/lichess-bot-devs/lichess-bot`), and the relevant code is in the lichess folder of our GitHub repo.

At the end of October 2024, after between 100 and 200 games per time format, AlphaGateau was able to reach a bullet Elo of 1991, a blitz Elo of 1829, and a rapid Elo of 1884.

**1. e4 c6 2. h3 h6 3. ♗e2 ♘a6 4. g4 g6 5. a4 ♖h7 6. f4 f6 7. ♘f3 ♗g7 8. h4 d5 9. e5 ♘b4 10. c3 ♗xg4 11. cxb4 ♗h5 12. ♘g1 ♗h8 13. ♘c3 d4 14. ♖h3 a5 15. exf6 ♘xf6 16. bxa5 e6 17. ♗xh5 gxh5 18. b4 ♘g8 19. ♖d3 ♘f6 20. ♘ge2 ♘g4 21. ♕c2 ♕xh4+ 22. ♘g3 ♕h2 23. ♘ge4 ♕h1+ 24. ♔e2 ♖g7 25. ♖xd4 ♕h2+ 26. ♔f3 ♕h1+ 27. ♔e2 ♖f7 28. ♖d3 ♕g2+ 29. ♔e1 ♖g7 30. ♖a2 ♕h1+ 31. ♔e2 ♔e7 32. ♖g3 ♖e8 33. ♕d3 ♖d8 34. ♕c4 ♖h4 35. ♖a1 ♖f7 36. ♖a2 ♖xf4 37. d4 ♖f7 38. ♗e3 ♖g7 39. ♖c2 ♔f7 40. ♕c5 ♖e8 41. ♖f3 ♕h2+ 42. ♔d3 ♕h1 43. ♖f4 ♘xe3 44. ♔xe3 ♖g1 45. ♖e2 ♖xd4 46. ♘f2 ♖xf4 47. ♘xh1 ♖h4 48. ♘f2 ♗d7 49. b5 ♖g3+ 50. ♔d2 ♗xc3+ 51. ♔c2 ♗h8 52. ♕b6 ♔d6 53. a6 bxa6 54. ♕d8+ ♔c5 55. ♕xh4 ♖g8 56. bxc6 ♗xc6 57. ♕f4 ♗g7 58. ♖xe6+ ♔d5 59. ♕f5+ ♔c4 60. ♕f3 ♔b4 61. ♕xh5 ♖xa4 62. ♖xa6+ ♔b4 63. ♕f5 ♖h8 64. ♖a7 ♗d4 65. ♖b7+ ♗b6 66. ♘d3+ ♔a4 67. ♖e7 h5 68. ♕f6 ♗d8 69. ♕f4+ ♔b5 70. ♖e5+ ♔b6 71. ♘f2 ♔c7 72. ♘e4 ♔c6 73. ♔c3 ♔b6 74. ♔d4 ♔c6 75. ♘g3 ♔b6 76. ♔c4 ♔b7 77. ♕f3+ ♔c7 78. ♔d5 h4 79. ♘e2 h3 80. ♕f4 ♔d7 81. ♘g3 ♔c7 82. ♖h5+ ♔d7 83. ♖xh8 h2 84. ♖xh2 ♗c7 85. ♕f2 ♔c8 86. ♕f5+ ♔b7 87. ♕d7 ♔a8 88. ♔e4 ♗b8 89. ♖h6 ♔a7 90. ♖f6 ♗g1 91. ♖f5 ♔b8 92. ♖f8#**

**Figure 15:** Game played by a model fully trained only on $5 \times 5$ chess as white. White is able to use their white bishop to eliminate black's white bishop, but seems to undervalue their knight on move 14, probably because it is a worse piece in $5 \times 5$ chess due to being more constrained and harder to effectively employ

**1. e4 c5 2. ♘b1c3 e6 3. ♘g1f3 ♘b8c6 4. d4 d4 5. ♘f3d4 ♘g8f6 6. ♘d4c6 c6 7. e5 ♘f6d5 8. ♘c3e4 ♕d8c7 9. f4 ♕c7b6 10. ♗f1e2 ♗c8a6 11. ♗e2a6 ♕b6a6 12. a3 h5 13. ♕d1e2 ♕a6e2 14. ♔e1e2 f5 15. ♘e4c3 a5 16. ♘c3d5 d5 17. ♗c1e3 ♗f8e7 18. c3 ♖h8g8 19. ♔e2f3 g5 20. g3 g4 21. ♔f3e2 h4 22. b4 ♔e8f7 23. ♖a1c1 h3 24. ♗e3b6 b4 25. b4 ♖a8a3 26. ♖c1c7 ♖g8b8 27. ♗b6c5 ♗e7c5 28. ♖c7c5 ♖b8b4 29. ♖c5c2 d4 30. ♖h1d1 ♔f7e7 31. ♔e2f2 ♖a3f3 32. ♔f2g1 d3 33. ♖c2c3 ♖b4b2 34. ♖c3d3 ♖f3d3 35. ♖d1d3 ♖b2c2 36. ♖d3d1 ♖c2g2 37. ♔g1h1 ♖g2a2 38. ♔h1g1 ♔e7e8 39. ♖d1b1 ♖a2g2 40. ♔g1h1 ♖g2c2 41. ♔h1g1 ♖c2c3 42. ♖b1b8 ♔e8e7 43. ♖b8b1 ♖c3c2 44. ♖b1b7 ♖c2c1 45. ♔g1f2 ♖c1h1 46. ♔f2e2 ♖h1h2 47. ♔e2f1 ♖h2g2 48. ♖b7b3 ♖g2d2 49. ♔f1g1 ♔e7f7 50. ♖b3b7 ♔f7f8 51. ♖b7b8 ♔f8f7 52. ♔g1h1 ♖d2g2 53. ♖b8b7 ♔f7e7 54. ♖b7d7 ♔e7e8 55. ♖d7d3 ♖g2f2 56. ♖d3d6 ♔e8e7 57. ♖d6c6 ♖f2f3 58. ♖c6c7 ♔e7d8 59. ♖c7c6 ♔d8d7 60. ♖c6d6 ♔d7e7 61. ♔h1h2 ♖f3f2 62. ♔h2h1 ♖f2f3 63. ♔h1h2 ♖f3f2 64. ♔h2h1 ♖f2g2 65. ♖d6d3 ♖g2c2 66. ♖d3d4 ♖c2c3 67. ♔h1h2 ♖c3a3 68. ♖d4d2 ♖a3a6 69. ♖d2e2 ♖a6a5 70. ♖e2d2 ♖a5a1 71. ♖d2b2 ♖a1a8 72. ♔h2h1 ♖a8c8 73. ♔h1g1 ♖c8a8 74. ♖b2b6 ♔e7d7 75. ♖b6d6 ♔d7e7 76. ♔g1h1 ♖a8a3 77. ♔h1h2 ♖a3a2 78. ♔h2h1 ♖a2a7 79. ♔h1g1 ♖a7a1 80. ♔g1h2 ♖a1a2 81. ♔h2h1 ♖a2e2 82. ♖d6d4 ♖e2g2 83. ♖d4d3 ♔e7e8 84. ♖d3a3 ♖g2e2 85. ♖a3d3 ♖e2e1 86. ♔h1h2 ♖e1e2 87. ♔h2h1 ♖e2b2 88. ♖d3c3 ♖b2a2 89. ♖c3c6 ♔e8d7 90. ♖c6d6 ♔d7e7 91. ♖d6c6 ♖a2e2 92. ♖c6c3 ♖e2g2 93. ♖c3d3**

**Figure 16:** AlphaGateau starts with a closed Sicilian, transposing into the Four Knights Sicilian, following a popular line until move 10, when white moves its white bishop to e2. The pawn structure locks the situation by move 35. Nothing much happens before the game ends in a draw, besides an interesting stalemate trick on move 54

**1. e4 e5 2. ♘f3 ♘c6 3. ♗b5 ♘f6 4. d3 ♗c5 5. ♗xc6 dxc6 6. O-O ♕e7 7. ♗g5 O-O 8. ♗h4 h6 9. ♘bd2 b5 10. ♕e1 a5 11. h3 ♗b6 12. ♗g3 ♖e8 13. ♗xe5 a4 14. ♗c3 ♘h5 15. a3 ♘f4 16. ♔h2 f5 17. ♕d1 ♖f8 18. exf5 ♖xf5 19. ♕e1 ♕f7 20. g4 ♖c5 21. ♘e4 ♖d5 22. ♖g1 ♘e6 23. ♘h4 ♘g5 24. ♘f6+ gxf6 25. f4 ♘e6 26. ♖g3 ♘d4 27. ♕e4 ♗d7 28. ♖e1 ♖e8 29. ♕g2 ♖xe1 30. ♗xe1 ♕e6 31. ♗f2 ♘e2 32. ♗xb6 cxb6 33. f5 ♕e5 34. ♘f3 ♕xg3+ 35. ♕xg3 ♘xg3 36. ♔xg3 ♗c8 37. ♔f4 ♖d7 38. ♘d2 ♔f7 39. ♘e4 ♗a6 40. h4 c5 41. g5 fxg5+ 42. hxg5 hxg5+ 43. ♘xg5+ ♔g7 44. ♔e5 ♖e7+ 45. ♘e6+ ♔f7 46. ♔d6 ♗c8 47. ♘g5+ ♔f6 48. ♘e4+ ♔f7 49. ♘g5+ ♔f8 50. f6 ♖d7+ 51. ♔c6 c4 52. ♘e6+ ♔f7 53. ♘f4 ♔xf6 54. ♘d5+ ♔e6 55. ♘xb6 cxd3 56. cxd3 ♗b7+ 57. ♔xb5 ♖xd3 58. ♘xa4 ♔d6 59. ♔c4 ♖h3 60. ♘c3 ♔c6 61. b4 ♗c8 62. a4 ♗e6+ 63. ♔d4 ♗b3 64. a5 ♖h4+ 65. ♔e5 ♖xb4 66. a6 ♔b6 67. ♘b1 ♗c2 68. ♘c3 ♗b3 69. ♘e2 ♔c5 70. ♘f4 ♖a4 71. ♘d3+ ♔c6 72. ♘f4 ♗c4 73. ♘g6 ♗xa6 74. ♘f4 ♗c4 75. ♘g2 ♗b3 76. ♘f4 ♗c2 77. ♘e6 ♖e4+ 78. ♔f5 ♖e1 79. ♔f6 ♔d6 80. ♘f4 ♖g1 81. ♘e2 ♖f1 82. ♔g5 ♔e5 83. ♘g3 ♖f7 84. ♘h5 ♗d3 85. ♘g3 ♖g7+ 86. ♔h4 ♖g8 87. ♔h3 ♔f4 88. ♘h5+ ♔g5 89. ♘g3 ♖h8+ 90. ♔g2 ♔f4 91. ♘f1 ♖b8 92. ♘g3 ♖b2+ 93. ♔h3 ♗g6 94. ♘h5+ ♗xh5 95. ♔h4 ♖h2#**

**Figure 17:** AlphaGateau starts playing a Berlin defence, without any book, and diverts by move 4 into a popular line, with a rare bishop move on move 7. The middlegame revolves around black strong knight, until white is forced to give up its remaining rook to stop the attack, leaving black up a rook for a pawn.

