# OpenReview forum: "Enhancing Chess Reinforcement Learning with Graph Representation"
_NeurIPS.cc/2024/Conference — NeurIPS 2024 poster_

### Official Review · Reviewer_abiC · 2024-07-03

**Soundness:** 1
**Presentation:** 3
**Contribution:** 3
**Rating:** 5
**Confidence:** 4

**Summary:**

The paper introduces a new variant of AlphaZero, AlphaGateau, with a neural network architecture based on GNNs, that allows the model to generalize across board sizes. The new architecture outperforms the original AlphaZero architecture under certain conditions.

**Strengths:**

The paper follows a promising direction, using GNNs in an AlphaZero-like model to utilize the graph nature of many board games.
AlphaGateau is an original model that shows potential in terms of data efficiency and fast training. It learns much faster than AlphaZero during the first 100 training iterations, saturating at a much higher Elo compared to that of AlphaZero, which does not improve much in this timeframe.
It is clear that when both models are tested on equal grounds, in the setting used, AlphaGateau is significantly more data efficient, and improves its Elo score much faster than AlphaZero. These are significant improvements on the original AlphaZero model if they hold under rigorous evaluation.

**Weaknesses:**

Evaluation & comparison with AlphaZero:
While figure 5 clearly shows that AlphaGateau learns much faster than AlphaZero at early training, the paper does not provide any information on how the two models compare when they are fully trained. AlphaGateau seems to reach a performance plateau within 100 steps, but the AlphaZero model trained by the authors would likely keep on improving for orders of magnitude more training steps. For comparison, the authors used a model size roughly 10% of Silver et. al.'s AlphaZero, but trained for 100 steps compared to the original 700,000 steps. It is not clear if AlphaGateau's performance will be comparable with AlphaZero, given enough training steps.
I am aware that training the models for 10^5 steps is extremely expensive and not feasible for this paper. Even so, the correct solution would have been to fit the model size to the training budget and train a significanly smaller AlphaZero model, for longer time and on smaller batches (or to heavily reuse training data between optimization steps). Doing a comparison of smaller-scale models could show if AlphaGateau is a comparable model to AlphaZero.

In the paper's current form, the reported Elo gap does not give the reader any knowledge about how AlphaGateau compares to a fully-trained AlphaZero, since Elo calculation is unreliable when using such large gaps. For example, the gap between the best performing AlphaGateau and best AlphaZero is about 1,500 Elo, meaning that AlphaZero would win about 1 game in every 7,500 games. This implies that the AlphaGateau Elo score is based purely on comparisons with earlier AlphaGateau training checkpoints.
Another example is the gap between the first and second checkpoints, which is hard to measure by eye due to the plot format but looks roughly equal to 800. This gap implies the weaker model wins 1 game in 100, suggesting that the error of this gap is very high unless the authors use thousands of test games between this specific checkpoint pair to calculate Elo. I couldn't find the number of Elo games in the paper, and would recommend stating it clearly.

Considering that the best model's Elo is probably based purely on comparisons to its earlier versions, and considering the possible error of the Elo estimate, it is possible that AlphaGateau's Elo rating will drop significantly if fully-trained AlphaZero models are added to the pool of players.

This paper will benefit significantly from fitting the size of the experiments to the compute resources available to the authors. I would suggest training significanly smaller models for 100x more training steps, using the same compute budget. In its current form, the paper only showcases the advantage AlphaGateau has in terms of data-efficiency at early training, in a regime where AlphaZero is severely undertrained.

**Questions:**

- What hyperparameters are used to calculate Elo scores? (number of models tested, number of games between each model pair, who played against who)

Three minor suggestions:
- It would be helpful for the reader if figures 5 and 6 had FLOPs on the x-axis instead of steps, or if training compute was reported. Plotting only steps, it is hard to understand if one training step of AlphaGateau is comparable to a training step of AlphaZero in terms of compute.
- It would be easier and more convincing to use traditional methods for calculating Elo scores, such as BayesElo which was used by Silver et. al. That said, the authors provide clear explanations on their Elo calculation method.
- The correct spelling of the metric is "Elo", named after Arpad Elo, rather than "ELO", which is a common typo.

**Limitations:**

No limitations other than those specified above regarding evaluation.

---

> ### Author Rebuttal · Authors · 2024-08-05
>
> # Remarks
>
> > the paper does not provide any information on how the two models compare when they are fully trained. AlphaGateau seems to reach a performance plateau within 100 steps, but the AlphaZero model trained by the authors would likely keep on improving for orders of magnitude more training steps
>
> Initial tests on the AlphaZero implementation of PGX seemed to indicate that the model converges in less than 400 steps, with marginal improvements after 100 steps, which is why we limited our experiments to 100 steps. However, we didn't test the runs in the paper for more than 100 steps. We started new longer runs on the AlphaZero model, to include either in the main paper or in the appendix.
>
>
> > For comparison, the authors used a model size roughly 10% of Silver et. al.'s AlphaZero, but trained for 100 steps compared to the original 700,000 steps. It is not clear if AlphaGateau's performance will be comparable with AlphaZero, given enough training steps.
>
> > This paper will benefit significantly from fitting the size of the experiments to the compute resources available to the authors. I would suggest training significanly smaller models for 100x more training steps, using the same compute budget. In its current form, the paper only showcases the advantage AlphaGateau has in terms of data-efficiency at early training, in a regime where AlphaZero is severely undertrained.
>
> In Silver et. al., they trained for 700,000 steps, however, due to their hardware availability, they were using 5000 TPUs to generate games, and 64 TPUs to train the network. As such, they could have the TPUs in charge of generating the games be always on and constantly generate games, while the TPUs in charge of training could just grab a random batch of positions from the last 1,000,000 generated positions to do one "step" of gradient update.
>
> In contrast, for our experiments, we used the same 8 GPUs alternating between generating data and training the neural network. When training, we sample as many batches as required without replacement as to fully cover the current frame window. What we call in the paper one step are the generation of around 130,000 positions, followed by around 500 to 1000 batches to cover the full frame window (depending on the size of the frame window we used).
>
> As such, our 100 steps are comprised of up to 100 000 batches. This is still an order of magnitude less than Silver et. al., but our model is also around an order of magnitude smaller. We will update the paper in order to clear up this confusing choice of term on our part.
>
>
> > or to heavily reuse training data between optimization steps
>
> As mentioned in the previous answer, we used a frame window, such that on average, each data point is reused around 7 times. We mentioned with Figure 7 part of our experiments on the impact of reusing training data, by varying the amount of data generated, and the amount of data reuse.
>
>
> > In the paper's current form, the reported Elo gap does not give the reader any knowledge about how AlphaGateau compares to a fully-trained AlphaZero, since Elo calculation is unreliable when using such large gaps.
>
> Each model is initially assumed to have an Elo of 1000, and plays 5 matches of 60 games each against 5 opponents. After each match the Elo of all the models is recomputed, and each opponent is chosen to be as close as possible to the current Elo evaluation of the model. Models with high difference in Elo are effectively not appropriate to evaluate the Elo ratings, but this is reflected in the plotted confidence intervals. We included in the global PDF response a distribution plot of all the Elo of the models that are included in the linear regression described in Eq (11), which shows that the ratings are quite continuous, without leaving any gaps that could bias the rating estimation. Each player played at least 300 games (150 as white and 150 as black), which was enough to keep the confidence intervals smaller than 100 Elo points.
>
>
> # Questions
>
> > What hyper-parameters are used to calculate Elo scores? (number of models tested, number of games between each model pair, who played against who)
>
> At the time of submission, we evaluated 255 players, and have since evaluated 977 players (pair of (model, parameters)). Each player played at least 60 games against each of 5 other players that were already evaluated (such that the graph of matches is 5-edge-connected). As the opponents of each player depended on their successive rating evaluations, they are not easy to list. However, the full  list of matches as well as their outcomes (wins, draws, losses) is included in the supplementary material zip in `rankings.json`
>
>
> > It would be helpful for the reader if figures 5 and 6 had FLOPs on the x-axis instead of steps, or if training compute was reported. Plotting only steps, it is hard to understand if one training step of AlphaGateau is comparable to a training step of AlphaZero in terms of compute.
>
> We do not easily have access to FLOPs data for our previous runs, however we do have access to timing data, which should loosely correlate with FLOPs as the same hardware was used for all experiments. We will include the corresponding plots in the appendix
>
> > It would be easier and more convincing to use traditional methods for calculating Elo scores, such as BayesElo which was used by Silver et. al. That said, the authors provide clear explanations on their Elo calculation method.
>
> Thanks, we were not aware of the existence of BayesElo. We compared the Elo from our method with the Elo for BayesElo (see plot in global PDF) and got relatively consistent results, besides the fact that weaker players were rated higher by our model, and similarly, stronger players were rater lower by our model, compressing the Elo range when compared to BayesElo. The main effective difference is a small amount of players were rated around 10 to 20 points out of sync with the players closest to them in Elo.

---

> ### Comment · Reviewer_abiC · 2024-08-07
> **AlphaZero Elo**
>
> Thank you for the detailed reply, I still need to go through all of it but wanted to clarify one point first. Reading the match-related numbers you specify, together with figure 1 in the PDF, I tend to believe your reported Elo scores are reliable, at least for the pool of AlphaGateau agents. It's also nice to see your Elo is not that different than that of BayesElo, making your results more robust.
>
> My problem with the comparison to AlphaZero still stands though. You claim (and I believe you) that your AlphaZero agents showed signs of saturation already after 100 'steps', meaning that according to figure 5 in the paper your AlphaZero models can only improve <300 Elo above an agent at 'step' 1, which is almost a purely random agent (right?). That means your AlphaZero still loses 15% of the time against an almost random agent, so clearly it failed to learn well.
>
> It is not necessary to train with a DeepMind-sized compute budget to get a good chess agent. One can train a small model tuned to the budget size to get very quick gains in Elo, that will saturate quicker the smaller the model is. I honestly don't know what went wrong with your training, you clearly have enough compute to go through a sufficient number of batches. Is the data-generating model only updated each 'step'? That could partially explain the problem, doing only 100 model updates is crucially little in my experience. Although it doesn't explain why the model would saturate so quickly. Could you specify how AlphaZero was trained? specifically how many games were played between optimization steps, how many batches used each optimization step, how often were the model weights updated (for the data-generation agent)?
>
> To summarize, I think your internal Elo rating is solid, but these Elo numbers are not anchored to any competent non-AlphaGateau agent, which makes it impossible to make any claim about AlphaGateau's performance.

---

> > ### Author Response · Authors · 2024-08-07
> >
> > With regards to our assessment of AlphaZero, we could have been clearer in the previous answer. It doesn't saturates after only 100 iterations, but around 400. 'with marginal improvements after 100 steps' was a poorly worded way of saying that the model stayed significantly worse than AlphaGateau, eventually reaching a rating between 300 and 500 Elo, compared to the more than 1500 of AlphaGateau. Seeing these results, we decided to only train AlphaGateau for 100 iterations as it seemed that the behavior of the model during these early iterations was the most important part, with both AlphaZero and AlphaGateau increasing significantly less afterwards.
> >
> > However, we do now agree with your criticism that the asymptotic behavior is also important to include. We will include in the revised version of the paper 500 steps of training of AlphaZero. We are currently running this experiment with our up-to-date code, with Elo ratings going up to iteration 300, where it still only goes from 100 elo at iteration 100 to around 450 by iteration 300.
> >
> > We will also note that AlphaGateau models don't saturate either by iteration 100. This was not clear in our initial figures as we only evaluated Elo rating once every 5 iterations, but it is clearer in our new figures evaluated every second iteration, such as the one in the global rebuttal pdf. We would further like to extend the runs of the AlphaGateau models to at least iteration 200, and will include those new runs in the revised paper.
> >
> > Yes, we only updated the data generating model once every step, and chose how many games to generate in one step to saturate our GPU. We could generate less games per step, but we would waste computing resources by doing so. We experimented trying to run more than one full epoch between each step of generation, but didn't notice any significant improvement behind the first 10 steps, so we decided to stick to only one epoch per step (and around 500-1000 batches) as the training already took the majority of our compute time.
> >
> > For the details of the training of AlphaZero, we generated 256 games at each step, and used batches of 2048 positions, for a total of 488 batches each step (except the first 6-7 steps, while the frame buffer is filling up). Each set of 256 generated games used newly updated network parameters. We also only used 128 MCTS simulations in each of our experiments.
> >
> > We didn't attempt to anchor our Elo ratings to other real agents, as our main results are the improvements of our architecture when compared to its AlphaZero basis. We however did let our latest fine-tuned for 20 steps 6-layer AlphaGateau model play around 600 blitz and bullet games on lichess, against mostly other bots, ending with an approximate Elo of 1800 in blitz, and 2000 in bullet. However, we had to adjust the number of MCTS simulations in order to make our model take an appropriate mostly constant amount of time to evaluate each move, which was often different to the number used during training, so these ratings are only indicative of the approximate abilities of that model.

---

> > > ### Comment · Reviewer_abiC · 2024-08-08
> > >
> > > Thank you for clarifying the details, this helps in understanding what went wrong with AlphaZero. Doing only 100 model updates is most probably the reason why it learns so slowly in figure 5. My guess is that 128 MCTS steps is also too small, given that chess has an average branching factor of 35. I honestly cannot tell you how to adapt your hardware for training with more model updates, I'm not familiar with the PGX implementation. But if there is a way you can reduce the quality of epochs in order to do x100 more model updates, that should probably lead to a good AlphaZero agent.
> > >
> > > I will change my score from reject to a borderline accept (5), for the following reasons:
> > > - This is a good paper considering anything other than evaluation.
> > > - The internal evaluation (tracking AlphaGateau's improvement against itself) in the PDF is a significant improvement on the one in the paper draft, and proves the model indeed improves fast during training.
> > > - The external evaluation (tracking AlphaGateau's improvement against AlphaZero) is not good. This is through no fault of the authors. Training AlphaZero is expensive and requires a lot of experience, the authors were just unlucky to choose bad training hyperparameters.
> > > - The results the authors report on Lichess increase the likelyhood that their models indeed learned well.
> > >
> > > I expect the authors to follow through with the changes they promised. Specifically, doing the following in the final version will minimize the damage of the evaluation issue:
> > > - Add a clear discussion of the limitations of their comparison to AlphaZero.
> > > - Train AlphaZero for longer and recalculating Elo for all agents. Hopefully this agent will improve and give more credibility to the results, but I cannot ask the authors to promise results they cannot control. If using BayesElo, I would suggest playing more matches with AlphaZero to increase these agents' influence on the final Elo scores.
> > > - Add the Lichess results to the paper or appendix. Using more MCTS steps is completely fine, especially given the low number used in training.
> > >
> > > I would have given this paper a much higher score if it could show significant improvement on AlphaZero, it's a shame that this paper suffers just because training the baseline is so hard.

---

### Official Review · Reviewer_R1p6 · 2024-07-08

**Soundness:** 3
**Presentation:** 3
**Contribution:** 2
**Rating:** 6
**Confidence:** 3

**Summary:**

The paper explores a novel approach to  reinforcement learning for Chess by utilizing a graph-based representation of the game state instead of the traditional grid-based representation. This method is based on GNNs  and aims to overcome the limitations of CNNs used in previous models like AlphaZero. Specifically, the paper introduces the Graph Attention neTwork with Edge features from Attention weight Updates (GATEAU) layer, which enhances the classical GAT layer by incorporating edge features.

The primary contributions of the paper include demonstrating that the new architecture outperforms previous models with a similar number of parameters and significantly accelerates the learning process. Additionally, the paper provides evidence that the model trained on a smaller variant of chess (5x5) can be quickly fine-tuned to perform well on the standard 8x8 chessboard, indicating promising generalization capabilities. The authors have made their code available, supporting the reproducibility of their experiments and findings.

**Strengths:**

**Novel Graph-based Approach**: This paper introduces a creative way to represent game states using a graph-based model instead of the traditional grid-based models. By employing Graph Neural Networks and the Graph Attention neTwork with Edge features from Attention weight Updates, the authors address significant limitations found in Convolutional Neural Networks used in models like AlphaZero.

**Improved Learning Efficiency**: The proposed AlphaGateau model demonstrates much faster learning compared to traditional CNN-based models. Experiments show that AlphaGateau can achieve a significant increase in playing strength in a fraction of the training time required by AlphaZero, which is a notable improvement in learning efficiency.

**Reproducibility**: The authors have made their code publicly available, supporting the reproducibility of their experiments.

**Weaknesses:**

**Limited Generalization Evidence**: The paper claims promising generalization capabilities, but the experimental evidence is limited to a small variant of chess (5x5) and standard chess (8x8). It would strengthen the paper to include additional experiments on other games or variants, such as Shogi or other board games with similar complexity, to demonstrate the broader applicability of the proposed architecture.

**Comparative Analysis**: The paper lacks a thorough comparative analysis with other state-of-the-art models beyond AlphaZero. Including comparisons with recent advancements in graph-based reinforcement learning models would provide a clearer picture of the relative performance and innovations of the proposed method.

**Real-world Applicability**: While the focus is on reinforcement learning in chess, discussing potential real-world applications of the proposed graph-based approach would broaden the impact of the work. Highlighting areas where this approach could be applied, such as other strategic games or decision-making problems in different domains, would make the contributions more compelling.

**Questions:**

1. How does AlphaGateau compare to other graph neural network-based reinforcement learning models, such as those presented by Ben-Assayag and El-Yaniv (2021) [1]?
2. Have the authors considered applying AlphaGateau to other strategic games beyond chess? If so, what preliminary results or observations can the authors share?
3. What potential real-world applications besides chess do the authors envision for the graph-based approach proposed in this paper?

[1] Ben-Assayag, S., & El-Yaniv, R. (2021). Train on small, play the large: Scaling up board games with alphazero and gnn. *arXiv preprint arXiv:2107.08387*.

**Limitations:**

The authors have acknowledged some limitations of their work, such as the restricted computational resources which prevented the training of deeper networks, and their focus solely on chess without extending experiments to other games. They also discussed challenges related to the reproducibility of their results due to the non-deterministic nature of parallelized GPU code. However, the paper could benefit from a deeper discussion on the broader implications of these limitations, particularly how they might affect the generalizability and robustness of the proposed method. Regarding potential negative societal impacts, the paper does not explicitly address this aspect.

---

> ### Author Rebuttal · Authors · 2024-08-05
>
> > 1. How does AlphaGateau compare to other graph neural network-based reinforcement learning models, such as those presented by Ben-Assayag and El-Yaniv (2021) [1]?
>
> We are not aware of GNN-based RL models that can be applied to chess besides a slightly adapted version of ScalableAlphaZero (from Ben-Assayag and El-Yaniv (2021)). ScalableAlphaZero differs from AlphaZero mainly in the fact that it replaced the CNN layers with GIN (Graph Isomorphism Network) layers, using a chess king grid as the graph. This should mainly be less expressive than AZ as the convolution kernel cannot treat the 8 surrounding squares differently, as opposed to how a CNN convolution would. The main advantage of ScalableAlphaZero is that it is able to scale, which allow samples of differently sized board to be given during training and testing. This allows for training on scaled down variants of Gomoku and Othello.
>
> As such, we believe that ScalableAlphaZero would perform similarly to AlphaZero under the same constraints, or possibly a little worse, due to the more complex functions to evaluate, and lower expressivity. We were not able to perform a fine-tuning experiment with ScalableAlphaZero to compare with AlphaGateau yet as the ScalableAlphaZero code is not available, and the model would require substantial changes to adapt to the action space of chess being linked to the graph edges rather than the graph nodes. We did not compare AG's performances to ScalableAlphaZero's on Othello or Gomoku either as those games do not seem like they would benefit much from a graph based representation.
>
>
> > 2. Have the authors considered applying AlphaGateau to other strategic games beyond chess? If so, what preliminary results or observations can the authors share?
>
> We have considered applying AlphaGateau to Shogi and Risk. We didn't test Shogi yet as we were more familiar with Chess while lacking knowledge and intuition for Shogi, and so could more easily interpret the results and play patterns of the model in Chess. We plan to address Risk in the future, but several challenges must be solved first, including handling more than 2 players, randomness (which can be handled by improvements to AlphaZero such as DeepNash), hidden information, and more complex turns.
>
>
> > 3. What potential real-world applications besides chess do the authors envision for the graph-based approach proposed in this paper?
>
> We believe that these kind of methods help handling graph based tasks, and could potentially be used as a basis in this kind of adversarial setting to electric network optimizations, or traffic minimization, for example.

---

> > ### Comment · Reviewer_R1p6 · 2024-08-08
> >
> > Thanks for your rebuttal and I will maintain my score!

---

### Official Review · Reviewer_NRN3 · 2024-07-14

**Soundness:** 3
**Presentation:** 4
**Contribution:** 3
**Rating:** 7
**Confidence:** 4

**Summary:**

The authors demonstrate a GNN that works on chess and is amenable to generalization.

**Strengths:**

# originality

This is the first chess GNN approach that performs well that I'm aware of in the literature, and requires some invoations in the edge representation.

# quality

The results look good, although the computational limitation of the work mean that a larger test would be good to demonstrate that this model scales up.

# clarity

The paper is clear and readable.

# significance

This is a new result in chess AI and looks like it could matter to the larger RL community. I'm not sure this will impact much chess engines in practice though as the GNN is likely much harder to optimize and implement than CNNs or heuristics, the memory usage on the attention mechanism for example would make implementing this in a high performance search code-base difficult.

**Weaknesses:**

The authors only look at self-play for model training, the original AlphaGo models were trained on human games to establish that the algorithm performed well, before starting the much more expensive self-play training. The lack of supervised training evaluations is makes the claims of strong performance limited to a single axis (sample efficiency).

This result mostly seems to arise from chess specific optimizations applied to an existing algorithm [18], while the authors suggest that it could be applied to other games they only provide evidence in chess.

The fine tuning results require more training time (and thus I'm assuming compute) than the initial training, this suggests the pre-training while beneficial is of limited efficacy unless it can be made more efficient.

I'm also concerned with the  attention-based pooling leading degrading in performance when the graph gets larger as chess is a game that cares much more about the max than the average.

I would have liked to see a more complete training run, but I understand that RL is very compute intensive so am not using this lack to negatively impact my score.

**Questions:**

Is the MCTS process identical between AlphaGateau and AlphaZero models? Or does the graph representation change it?

Could the graph representation be leveraged to allow a different search algorithm?

Could the authors provide PGNs with model predictions and some discussion of the difference in the AlphaGateau and AlphaZero models? There are patterns that CNN based models are good at learning and some that are more difficult (see [1] for examples) and showing differences in the models would strengthen this result.

Does the model handle the white bishop well when scaled up? If it does that would be additional evidence of generalization when scaling up since the moves would not have been seen before.

Can you provide more details on the Elo calculation? You use the terms ELO and elo and seem to refer to a varation of the Elo rating system created by Arpad Elo, but don't give much details beyond the optimizer. Are you using and Elo or Glicko (2) type system or something else? Section A has not citations so it's unclear.

[1] McGrath, T., Kapishnikov, A., Tomašev, N., Pearce, A., Wattenberg, M., Hassabis, D., Kim, B., Paquet, U. and Kramnik, V., 2022. Acquisition of chess knowledge in alphazero. Proceedings of the National Academy of Sciences, 119(47), p.e2206625119.

**Limitations:**

Discussed above

---

> ### Author Rebuttal · Authors · 2024-08-05
>
> # Remarks
>
> > The results look good, although the computational limitation of the work mean that a larger test would be good to demonstrate that this model scales up.
>
> We are aware that the reduced scale compared with the original AlphaZero is an issue, we have since ran a fine-tuning experiment with a model of depth 8 that will replace the model of depth 6 in the initial paper. The corresponding plot is included in the global PDF
>
>
> > the memory usage on the attention mechanism for example would make implementing this in a high performance search code-base difficult.
>
> We do not compute attention between each pair of node, but only when an edge is already present. As such, it shouldn't represent a significant memory sink, as there is only one attention coefficient per edge, which already have a feature vector of size 128.
>
>
> > This result mostly seems to arise from chess specific optimizations applied to an existing algorithm [18], while the authors suggest that it could be applied to other games they only provide evidence in chess.
>
> Yes, this paper only experimented on chess. It would be relatively straight-forward to extend the model to Shogi, as the rules are quite similar besides piece-dropping, which could provide insights to the capacity of the model to learn general game concepts if an hybrid chess/Shogi model was trained in some way.
> We also plan to look into extending this model to the game of Risk, which is even more suited to graph representations, but several challenges will need to be solved, as Risk has more than 2 players, hidden information, and randomness, which are not currently well handled by AlphaZero (although there exists extensions that focus on those issues, like DeepNash)
>
>
> > The fine tuning results require more training time (and thus I'm assuming compute) than the initial training, this suggests the pretraining while beneficial is of limited efficacy unless it can be made more efficient.
>
> The fine tuning on 8x8 chess is slower than the initial training on 5x5 chess, but the fine-tuning starts with a model that is around 1000 Elo point higher than a random initial model, which shows that it can transfer the learning from a smaller variant to the regular variant quickly. It still requires a significant amount of training to achieve optimal performances, but it should be possible to train a large model with strong hardware to provide a good baseline that can be used for further fine-tuning with more modest hardware on a wide range of chess variants, such as chess960, king of the hill, or other (in a similar way as how LLM models are used).
>
>
> > I'm also concerned with the attention-based pooling leading degrading in performance when the graph gets larger as chess is a game that cares much more about the max than the average.
>
> I'm not quite sure I understand the issue raised. In our experiments, the graph remains relatively small, containing 64 nodes for 8x8 chess and 1858 edges (25 nodes and 455 edges for 5x5 chess).
>
>
> > I would have liked to see a more complete training run, but I understand that RL is very compute intensive so am not using this lack to negatively impact my score.
>
> The original AlphaZero paper trained for 700 000 "steps". However, this is confusing as we also use the term step in our paper (and will clarify this point in the paper) but applied to something else. AlphaZero's steps are equivalent to batches in our paper, and we evaluate between 500 and 1000 batches in one of our step (depending on the size of the frame window). As such, our 100 steps are comprised of up to 100 000 batches. This is still an order of magnitude less, but our model is also around an order of magnitude smaller. We however are also working on a base AlphaZero run with 500 steps to assess whether it tapers off, or continues to increase in performance, after reading the reviews.
>
> # Questions
>
> > Is the MCTS process identical between AlphaGateau and AlphaZero models? Or does the graph representation change it?
>
> > Could the graph representation be leveraged to allow a different search algorithm?
>
> Yes, we use the same MCTS algorithm as for Gumbel MuZero, which uses the value and policy neural network oracle as a black box. We did not find a way to use the graph representation to improve it.
>
>
> > Could the authors provide PGNs with model predictions and some discussion of the difference in the AlphaGateau and AlphaZero models? There are patterns that CNN based models are good at learning and some that are more difficult (see [1] for examples) and showing differences in the models would strengthen this result.
>
> We did not analyse in detail the AlphaZero model as it performed quite poorly. For illustration, after 170 steps, the AlphaZero model distribution of first moves over 240 games as white is ` b2b3: 48, Ng1f3: 44, f2f3: 35, g2g4: 26, Nb1a3: 25, b2b4: 19, c2c4: 13, Ng1h3: 9, a2a4: 4, c2c3: 4, h2h3: 3, d2d4: 3, d2d3: 3, a2a3: 2, Nb1c3: 1, g2g3: 1`
>
>
> > Does the model handle the white bishop well when scaled up? If it does that would be additional evidence of generalization when scaling up since the moves would not have been seen before.
>
> The model seems to have a reasonable grasp of 8x8 chess, even when it was only trained on 5x5 positions. We can see from its distribution of first white move ` b2b3: 16, e2e4: 15, g2g3: 13, a2a3: 12, h2h3: 11, f2f4: 11, c2c3: 11, a2a4: 11, e2e3: 9, d2d3: 8, h2h4: 7, f2f3: 7, b2b4: 6, g2g4: 5, c2c4: 3, Ng1f3: 2, d2d4: 2, Nb1c3: 1`, that it is drawn to playing double pawn moves, which weren't available in 5x5 chess. We also included a PGN of a game it played as white where it was able to use and understand the white bishop, but also probably undervalued the knight as it was quite a restrained piece in 5x5
>
>
> > Can you provide more details on the Elo calculation?
>
> We use the base Elo rating model, but instead of updating the ratings after each game, we periodically run a linear regression following Eq (11) to fit Elo ratings on the players.

---

> > ### Comment · Reviewer_NRN3 · 2024-08-09
> >
> > Thank you for your detailed response. As I said in my review I am not judging the work by the amount of compute. I think this is an informative paper on GNNs for game play by itself and while I would like a more detailed examination of it's scaling with respect to other models I don't think that is a requirement for acceptance, but the other reviewers disagree so I will continue to watch the discussion. For now I maintain my score.
> >
> > Also to explain my point about scaling. I the GNN presented here does not use max pooling, instead it uses a weighted average (via attention). This will limit the depth the model can "see" important nodes since information about them will be lost as it passes up network. This is different from the PUCT algorithm which maintains information about the best lines even if they are very deep via counting the number of times each node has been searched and using that as its deciding factor instead of the approximated Q value which is an average. Thus I suspect that there will be scaling issues with this model as the networks increase in scale.

---

### Author Rebuttal · Authors · 2024-08-06

We thank the reviewer for their time and thoughtful opinions.

As shown by reviewer abiC, we were unclear in our terminology and cause confusion when compared to Silver et. al. (2017) with regards to what we called steps, leading to a first impression of our experiments being smaller than they actually were. We will clarify this point in our paper.

We have since the initial submission been able to train a deeper model, with 8 layers instead of 5 and 6, which further confirm our initial results, and included a revised version of Figure 6 in the joint pdf. We also retrained the base 5-layers model in order to be able to evaluate it every second iteration rather than the 1 in 5 it was initially. In addition, we found a bug in our implementation that made some of the models presented in the figure have slightly different hyper-parameters than reported. We have since re-run all affected models resulting in similar or slightly improved results. The paper will be updated with the new 8 and 5 layers run, as well as the fixed reruns.

We also plan to include in the appendix a version of each plot with `running time` on the x axis rather than `iteration`, as a proxy to FLOPs, following the recommendation of reviewer abiC.

A pgn of a game played as white on 8x8 chess by a model trained only on 5x5 chess was also included in the joint pdf, to show the model ability to learn general pattern and rules in a simplified version of the game. We will add more pgns to the appendix of the full paper.

We also recognize that our method of computing Elo ratings could have been better justified. We included in the joint pdf 2 plots justifying that the collection of models tested didn't leave gaps in the rating range, and that our method produced results consistent with ratings estimated using BayesElo.

We would be happy to engage in further discussion if you feel there is any remaining questions that we left unanswered, or if our previous answers were lacking in some way.

---

### Comment · Area_Chair_AHvk · 2024-08-07
**Discussion Period**

Hi all! Just your friendly area chair checking in.

First, thanks to the reviewers for your work so far on this paper. The discussion period has begun, which includes both reviewers and authors. This is our opportunity to clear up misunderstandings, get answers to questions, and generally gather enough information to make an informed decision grounded in the acceptance criteria.

This paper has a sizable spread of overall assessments from the reviewers. It seems there is general agreement that the approach in the paper is interesting and promising. The main critiques raised seem to concern lack of support for claims of generality, the size of the model and extent of training (which may not cover asymptotic performance comparisons), and the validity of the ELO ratings and comparisons. Does this seem like a fair summary?

Reviewers: please carefully read the other reviews and the authors' responses (with included PDF) and let us know what follow-up questions you have and to what extent your evaluation might change as a result of the additional context. Please especially raise any points of disagreement with other reviewers or the authors, as these are opportunities for us to clarify misunderstandings and detect misconceptions.

---

### Decision · Program_Chairs · 2024-09-25

**Decision:**

Accept (poster)

**Comment:**

The reviewers agree that the application of graph neural networks to chess is interesting and worth studying. The reviewers note some weaknesses in the evaluation of the method and the comparison to AlphaZero, but, during the discussion period, the most serious of the concerns were addressed, and the authors have offered additional results and clarifications that will strengthen the paper. Though the reviewers uniformly agree that the paper is acceptable, the somewhat limited scope/impact of the work and the limitations on the conclusions that can be drawn from the experimental evaluations suggest that this work is best presented as a poster, rather than as a spotlight or oral presentation.